# Identification of Key Transcription Factors Related to Bacterial Spot Resistance in Pepper through Regulatory Network Analyses

**DOI:** 10.3390/genes12091351

**Published:** 2021-08-29

**Authors:** Qingquan Zhu, Shenghua Gao, Wenli Zhang

**Affiliations:** 1State Key Laboratory for Crop Genetics and Germplasm Enhancement, Collaborative Innovation Center for Modern Crop Production Co-Sponsored by Province and Ministry (CIC-MCP), Nanjing Agricultural University, No.1 Weigang, Nanjing 210095, China; 2019101149@njau.edu.cn; 2Hubei Key Laboratory of Vegetable Germplasm Enhancement and Genetic Improvement, Cash Crops Research Institute, Hubei Academy of Agricultural Sciences, Wuhan 430070, China; shgao@webmail.hzau.edu.cn

**Keywords:** pepper, *Xcv*, RNA-seq, co-expression module, gene regulatory network

## Abstract

Bacterial spot (BS), caused by *Xanthomonas campestris* pv. *Vesicatoria* (*Xcv*), severely affects the quality and yield of pepper. Thus, breeding new pepper cultivars with enhanced resistance to BS can improve economic benefits for pepper production. Identification of BS resistance genes is an essential step to achieve this goal. However, very few BS resistance genes have been well characterized in pepper so far. In this study, we reanalyzed public multiple time points related to RNA-seq data sets from two pepper cultivars, the *Xcv*-susceptible cultivar ECW and the *Xcv*-resistant cultivar VI037601, post *Xcv* infection. We identified a total of 3568 differentially expressed genes (DEGs) between two cultivars post *Xcv* infection, which were mainly involved in some biological processes, such as Gene Ontology (GO) terms related to defense response to bacterium, immune system process, and regulation of defense response, etc. Through weighted gene co-expression network analysis (WGCNA), we identified 15 hub (Hub) transcription factor (TF) candidates in response to *Xcv* infection. We further selected 20 TFs from the gene regulatory network (GRN) potentially involved in *Xcv* resistance response. Finally, we predicted 4 TFs, C3H (p-coumarate 3-hydroxylase), ERF (ethylene-responsive element binding factor), TALE (three-amino-acid-loop-extension), and HSF (heat shock transcription factor), as key factors responsible for BS disease resistance in pepper. In conclusion, our study provides valuable resources for dissecting the underlying molecular mechanism responsible for *Xcv* resistance in pepper. Additionally, it also provides valuable references for mining transcriptomic data to identify key candidates for disease resistance in horticulture crops.

## 1. Introduction

Pepper (*Capsicum annuum* L.) is an important vegetable crop for our daily consumption around the world. It is rich in various nutrients beneficial to human health such as capsaicin, carotenoids, and vitamin C [1]. The pepper has a wide range of species and is a popular vegetable cultured all over the world [2]. However, the production of pepper is always affected by various biotic stresses such as bacterial spot, powdery mildew, and bacterial wilt [3], and abiotic stresses such as low temperature and drought [4,5]. Among these stresses, pepper bacterial spot (BS) disease, caused by *Xanthomonas campestris* pv. *Vesicatoria* (*Xcv*), is one of the most common bacterial diseases for pepper culture and is harmful to the yield and quality of pepper [6], therefore, resulting in huge economic losses to the producers.

*Xcv* was first discovered in tomato and termed as the pathogenic bacteria *Xcv* by Doidge in 1921 [7]. Subsequently, *Xcv* was found to cause bacterial spot on peppers [8]. *Xcv* usually infects stems, leaves, petals, and fruits of peppers, resulting in deciduous leaves, flowers, and fruits, and even the death of the whole plant [9]. Effective prevention and control of pepper BS disease usually include seed treatment, crop rotation, chemical sterilization, and selection or breeding of disease-resistant cultivars [10]. However, the breeding and application of disease-resistant cultivars is the most efficient and environment-friendly solution to prevent the problem of *Xcv* infection during pepper production [10]. To date, five dominant hypersensitive resistance (R) genes, *Bs1*~*Bs4* [10] and *Bs7* [11], and two non-hypersensitive recessive R genes, *bs5* and *bs6* [12], have already been identified in pepper. Most of them have been applied in the breeding of pepper resistance to *Xcv*. To provide more genetic resources for the breeding of pepper with enhanced resistance to *Xcv*, more genes with the potential to improve the resistance to BS disease in pepper need to be extensively explored. 

Currently, a large amount of RNA-seq data has been applied to identify candidate genes responsible for the improvement of crop yield and quality, as well as response to various stresses in plants [13,14,15,16,17]. Weighted gene co-expression network analysis (WGCNA) is a common systematic biology method for identifying hub (Hub) genes in the network through large-scale RNA-seq data [18]. This approach provides an effective way to mine Hub genes in response to various stresses in plants [19,20,21,22,23], however, it remains difficult to infer causality according to co-expression networks [24]. Tools such as ARACNE [25] and GENIE3 [26] attempt to construct gene regulatory networks (GRN) from co-expression networks. GENIE3 incorporates TF information to construct a regulatory network by determining the TF expression pattern that best explains the expression of each of their target genes [24]. This method has been widely applied to identify the regulatory relationships between TFs or TFs and their target genes in humans and plants [27,28,29,30]. Moreover, the low nitrogen response-related TFs and their GRN in rice were successfully predicted through combining WGCNA and GENIE3 [31], indicating the accuracy of these methods in predicting the gene regulatory relationships through constructing networks.

The differentially expressed genes (DEGs) in VI037601 and early Calwonder (ECW), representing resistance and susceptibility to *Xcv* infection, respectively, at 0 h, 6 h, and 24 h post *Xcv* infection, have already been characterized by Gao et al. [32]. However, it is still unclear how genes are differentially expressed between VI037601 and ECW at 0 h, 6 h, and 24 h post *Xcv* infection. Particularly, studies related to the identification of TFs and their regulatory target genes responsible for *Xcv* resistance through co-expression and regulatory networks are still missing in pepper. In this study, we re-analyzed public multiple time-point related RNA-seq data sets generated from the *Xcv*-susceptible cultivar ECW and the *Xcv*-resistant cultivar VI037601 post *Xcv* infection for the identification of DEGs that occurred between VI037601 and ECW at 0 h, 6 h, and 24 h post *Xcv* infection; we then conducted WGCNA and GRN analyses and identified several TFs potentially responsible for BS disease resistance in pepper. 

## 2. Materials and Methods

### 2.1. Re-Analyses of Public RNA-Seq Data Sets

Public RNA-seq data sets are available from NCBI’s SRA (Sequence Read Archive) database with accession number PRJNA693027 [32]. The detailed procedures for generating RNA-seq data from two bell pepper cultivars, VI037601 and ECW, representing resistance and susceptibility to *Xcv* infection, respectively, were previously described [32]. Briefly, two cultivars were grown at 25 °C/20 °C day/dark time and 16 h/8 h light/dark cycle with an approximately 60% relative humidity in standard glasshouse conditions. After growing on nutrient agar medium at 28 °C for two days, *Xcv* strain 23-1 was diluted to make a suspension with a concentration of 2 × 10^8^ cfu/mL using sterile water. Five leaves-stage plants of ECW and VI037601 were used for inoculation with procedures as below: the suspension was injected into the abaxial leaf surface near the midrib of the third to fifth pepper leaves using a syringe, the inoculated area was about 1.5–2 cm diameter. The inoculated leaf fragments were collected at 0 h, 6 h, and 24 h post-infection (hpi, hour post-infection), respectively. The collected samples were ground into a fine powder in liquid nitrogen either immediately used for RNA extraction and RT-qPCR assays or stored at −80 °C for later use. Fifteen leaves randomly selected from five different plants were polled as a biological replicate. Three independent biological replicates were prepared for each treatment.

For reanalysis of those RNA-seq data sets, fastq-dump software (v2.10.0) (https://ncbi.github.io/sra-tools/fastq-dump.html, accessed on 28 June 2021) was used to convert SRA data into fastq format data, then, FastQC software (v0.11.9) (https://www.bioinformatics.babra-ham.ac.uk/projects/fastqc/, accessed on 28 June 2021) was used to evaluate the data quality. The adapters and low-quality reads were removed by fastp software (v0.21.0) [33], and retained clean reads were used for further analysis. 

### 2.2. Identification of DEGs

The filtered clean reads were mapped to the reference genome of cultivated pepper “Zunla-1” using Hiast2 software (v2.1.0) [34]. Gene expression was quantified using featureCounts software (v1.6.4) [35] followed by DEGs at three time points post *Xcv* infection between two cultivars using the R package DESeq2 [36]. The threshold of DEGs was set to |log_2_(FoldChange)| ≥ 1 and the false discovery rate (FDR) values < 0.05. Fragments per kilobase of transcript per million fragments mapped read (FPKM) values were calculated based on the length of genes and the read counts mapped to genes. 

### 2.3. qRT-PCR Assay

The qRT-PCR assay was conducted following the published procedures [32]. Briefly, the first cDNA strand was reverse-transcribed from total RNA extracted from *Xcv* infected leaves of ECW and VI037601, which is similar to the RNA used for RNA-seq, using the kit purchased from TransGen (Beijing, China). qRT-PCR was conducted using the SYBR Premix Ex Taq^TM^ kit (Takara, Dalian, China) on the Biosystems QuantStudio 5 Real-Time PCR system (Applied Biosystem, Waltham, MA, USA, USA). CaUbi3 (AY486137.1), encoding ubiquitin-conjugating protein, was used as an internal control. Relative expression levels of genes examined were calculated from three independent biological replicates and expressed as 2^−^^ΔΔ^^CT^. Primers for the qRT-PCR assay are listed in Appendix A.

### 2.4. Co-Expression Network Analysis

Weighted gene co-expression network analysis was performed using the R package WGCNA (v1.69) [18]. The FPKM matrix of gene expression was used as input data, the genes with low variation of expression levels among 18 samples were filtered out according to the median absolute deviation (MAD) value > 0.5 of each gene, and the remaining 12,442 genes were used to construct the network. By using the function pickSoftThreshold in the R package, the best-weighted coefficient was determined as β = 7, and the scale-free network was constructed according to the default parameters. The threshold used to screen module Hub genes was |kME| > 0.85.

### 2.5. Construction of GRN 

The R package GENIE3 (v1.12.0) [26] was used for the inference of GRN. The input expression matrix was the same as that used in WGCNA. According to the results of the PlantTFDB database [37], 661 of 12,442 genes encoded TFs. The 661 genes were designated as potential regulatory factors in the processes of machine learning. The top 10,000 predicted regulatory relationships were extracted from GRN as potential regulatory networks based on the calculated “weight” values (≥0, a measure of the regulatory confidence level as defined by GENIE3). The 20 TFs with the heaviest summed weights were chosen as candidates for GRN. All networks were visualized using Cytoscape (v3.6.1) software [38]. The details of R codes for WGCNA and GENIE3 are provided in Method S1.

### 2.6. Gene Ontology Annotation and Enrichment Analysis

Gene Ontology (GO) annotation was conducted using the eggNOG [39]. Functional enrichment analysis was performed by the R package clusterProfiler (v3.14.3) [40] with *p.adjust* < 0.05. Redundant GO terms were filtered out by using the REVIGO website [41].

### 2.7. Prediction of Cis-Regulatory Elements for Candidate TF Target Genes

For motif enrichment analysis, the 1000 bp upstream of the transcription start sites (TSSs) of candidate TF target genes were used for the enrichment of known cis-elements using AME [42]. The top three most significant enriched motifs are shown. 

## 3. Results

### 3.1. Identification of DEGs Post Xcv Infection between Two Pepper Cultivars 

To investigate the dynamic changes of gene transcription responding to *Xcv* infection between the susceptible pepper cultivar ECW and the resistant pepper cultivar VI037601, we extracted 18 RNA-seq samples and performed principal component analysis (PCA) by the FPKM expression. As illustrated in Figure 1A, the biological replicates of each sample were well correlated; transcriptional changes occurred in each cultivar post *Xcv* infection. Moreover, there were obvious differences between the two cultivars at the same infection time point, which also demonstrated the different resistance between the two cultivars in response to *Xcv* infection. 

Next, we identified a total of 3568 DEGs in VI037601 relative to ECW at the same time point post *Xcv* infection (Figure 1B). The details for those DEGs are listed in Appendix A. Among three time points, the greatest number of DEGs was detected at 6 hpi, including 1918 up-regulated genes and 341 down-regulated genes, the second greatest number of DEGs occurred at 24 hpi, including 1033 up-regulated genes and 782 down-regulated genes. Moreover, the number of up-regulated genes at each time point was much more as compared to the down-regulated genes. According to pair-wise comparison (Figure 1C), we found that 298 DEGs were common at all three time points post *Xcv* infection, while 295, 1181, and 960 DEGs specifically occurred at 0 hpi, 6 hpi, and 24 hpi, respectively. Thus, all the above analyses showed that *Xcv* infection resulted in dramatic transcriptional changes between two cultivars.

To assess if those DEGs have any biological relevance in response to *Xcv* infection, we performed GO functional enrichment analyses for DEGs corresponding to each time point (Appendix A). Redundant GO terms were filtered out due to the excessive number of enriched GO terms (Figure 2). Some distinct or common biological processes were observed at individual or all time points post *Xcv* infection. For example, GO terms associated with response to wounding, secondary metabolic process, and secondary metabolite biosynthetic process were enriched at all time points; GO terms related to response to chitin, regulation of defense response, defense response to bacterium, and immune system process were detected at 0 and 6 hpi. Moreover, GO terms related to lipid transport and negative regulation of leaf development were specifically enriched at 6 hpi, while DEGs at 24 hpi were specifically associated with biological processes such as regulation of defense response to bacterium, nucleosome assembly, and nitrate transport, etc. 

In addition, we observed differential enrichment of GO terms associated with response to bacterium between 6 hpi and 24 hpi. A plausible explanation is the following: 6 hpi may be the key time point for VI037601 and ECW responding to *Xcv* infection through up- or down-regulating some of the genes. Those genes can function in defense response and in response to antibiotics, bacteria, chitin, hypoxia, wounding, acid chemical, and drug for enhancing resistance to *Xcv* infection, thereby reducing disease-related responses and promoting the recovery of plants for normal growth to some extent at 24 hpi. This process may cause differential enrichment of GO terms associated with stress or bacterium responses at 24 hpi relative to 6 hpi. 

Collectively, all of the above analyses show that DEGs are responsible for differential responses to *Xcv* infection between two cultivars.

### 3.2. Identification of Co-Expression Modules in Relation to Xcv Resistance Response 

To explore the genes that play key roles in *Xcv* resistance response in pepper, we used WGCNA to construct gene co-expression modules. All genes in the same module have similar expression patterns in different samples. A total of 18 co-expression modules were identified (Figure 3A), modules were distinguished by different colors. The grey module contained the genes (n = 20) that were not effectively clustered. The number of genes in the remaining modules ranged from 30 (grey60) to 7070 (turquoise) (Appendix A). We calculated module eigengene values (ME, defined as the first principal component of genes in a module, represents the overall expression pattern of the module) for each module in different samples. According to the ME values, when compared with the ECW cultivar, we found that at three time points post *Xcv* infection, the eigengene expression levels in the black module were significantly lower in the VI037601 cultivar (Figure 3B), indicating that gene expression in this module was consistently suppressed in the VI037601 cultivar, while the eigengene expression levels in module greenyellow and pink were significantly higher in the VI037601 cultivar (Figure 3C,D). Although the genes in these three modules were not significantly enriched in any GO functions, we found that the GO annotations of many genes in these modules were related to immunity, defense response, abiotic stress, bacterial response, and other functions related to pathogen invasion (Appendix A). We eventually determined to use these three modules as the "Pepper *Xcv* resistance response modules" for subsequent analyses.

### 3.3. Identification of Hub Genes Related to Disease Resistance Response 

To identify Hub genes involved in *Xcv* resistance response, we further screened these three *Xcv* resistance response modules using the module membership (kME) values calculated by the R package, the size of |kME| representing the core degree of a gene in a module. We indeed identified multiple Hub genes in each module (Appendix A). The transcription factor plays an important role in different physiological and biochemical processes during plant–pathogen interactions [43]. Therefore, we further identified a total of 15 Hub TFs out of all Hub genes, which belong to B3 (third basic domain), C3H (p-coumarate 3-hydroxylase), CO-like (CONSTANS-like), TALE (three-amino-acid-loop-extension), HSF (heat shock transcription factor), ERF (ethylene-responsive element binding factor), and other TF families (Figure 4). As illustrated in Figure 4, TFs were interconnected with each other forming a complicated network, suggesting that these TFs might play important regulatory roles in pepper *Xcv* resistance response. 

### 3.4. Construction of GRN by Machine-Learning for Identification of Key TFs

WGCNA is an effective method for indicating co-expression relationships among genes and identifying some Hub genes in the network [24]. However, it is not easy to predict regulatory relationships among genes. To examine the regulatory relationships between TFs or between TFs and their target genes, we further employed a machine learning approach to infer regulatory causality for co-expressed genes. 

We selected 20 TFs with the most regulatory potential and constructed their transcriptional regulatory network (Appendix A). Given that not all TF genes listed in Appendix A were inducible in response to *Xcv* infection, thus GRN analysis may help to identify key TFs involved in *Xcv* infection response. After associating these 20 TFs with 15 Hub TFs presented in the aforementioned modules related to *Xcv* resistance response, we finally obtained 4 TFs, Capana00g004634 (ERF family), Capana09g000015 (C3H family), Capana11g000364 (TALE family), and Capana03g003818 (HSF family), as candidate genes related to *Xcv* resistance in pepper. These 4 TFs were common genes in the Hub TFs of WGCNA and the 20 most potential TFs of GRN. Their homologous genes in *Arabidopsis thaliana* are listed in Appendix A.

We further extracted the regulatory sub-network containing these 4 TFs from the entire GRN of 20 TFs (Figure 5A). It contained 195 genes for module black, greenyellow, and pink accounting for approximately 71% of the total 276 genes from the entire regulatory sub-network. As illustrated in Figure 5B, substantial target genes were observed to be associated with 4 TFs, especially for Capana09g000015 and Capana11g000364. Interestingly, these 4 TFs were also predicted to have certain mutual regulatory relationships (Figure 5C). For instance, the HSF family gene in the pink module can potentially regulate the C3H family gene in the black module, and the 3 TFs in the black module were predicted to regulate each other, indicating the complex regulatory relationships among these key TFs in response to *Xcv* infection.

Moreover, 41 genes were predicted to be co-regulated by the 3 TFs in the black module (Figure 5C), while the majority of them were Hub genes that did not encode TFs (Appendix A) in the three modules related to *Xcv* resistance response, such as Capana07g001550 and Capana02g001862, suggesting that these TFs in responding to *Xcv* infection were mediated by some Hub target genes related to *Xcv* resistance response. The GO function of these target genes was involved in the regulation of secretion. Surprisingly, among the aforementioned 41 genes co-regulated by the 3 TFs in the black module, 40 genes belong to the three modules related to *Xcv* resistance response; there were 17 genes that were co-regulated by all those 4 TFs (Appendix A).

### 3.5. Validation of TF Candidates

To access if 4 TF candidates can regulate the expression of their target genes through direct binding, we conducted cis-element enrichment analyses for 1000 bp upstream of TSS (putative promoter regions) of 4 TF target genes using AME [42]. We detected that CCTCCT-repeat and CTTTTT motifs were enriched in the promoters examined (Table 1). It has been documented that CCTCCT-repeat is the binding sequence for the C2H2 TF family TFIIIA in *Arabidopsis*. TFIIIA and L5 in *Arabidopsis* exhibit a high affinity for binding potato spindle tuber viroid RNA in vitro [44]. Moreover, the CTTTTT motif can be recognized by OBP3 responsive to salicylic acid in *Arabidopsis* [45]. Thus, the cis-element enrichment analyses indicate the potential regulatory relationships that occurred between the 4 TF candidates, identified using the machine-learning approach, and their target genes through the network.

To validate the accuracy of RNA-seq data, we selected 12 key genes (Appendix A) in the co-expression network or GRN for qRT-PCR assay. They included 4 TF candidates (Capana00g004634, Capana09g000015, Capana11g000364, and Capana03g003818) identified in our study, two TFs (Capana08g001044 and Capana06g000455) in the top 20 TFs identified by GRN but not shown up in our study, and two Hub TFs genes (Capana08g002479 and Capana04g000509) belonging to members of 15 Hub TFs in WGCNA and the remaining 4 Hub genes selected from the co-expression network. As shown in Figure 6, the expression profiles of those 12 genes examined by qRT-PCR exhibited an overall similar trend with RNA-seq data. For qRT-PCR in VI037601 post *Xcv* infection, we found that 7 genes (Capana00g004634, Capana03g003818, Capana08g001044, Capana08g002479, Capana06g000455, Capana04g000509, and Capana06g000045) were expressed more at 6 hpi than the other two time points, 3 genes (Capana11g000364, Capana01g004363 and Capana00g000428) were up-regulated at 6 hpi and 24 hpi, and 2 genes (Capana09g000015 and Capana01g000810) tended to be down-regulated at 6 hpi and 24 hpi. In particular, up-regulated Capana01g004363 and Capana00g000428 are the target genes of these TF candidates, indicating the positive roles of TF candidates in regulating their target genes. In short, qRT-PCR results validate RNA-seq analyses in our study.

## 4. Discussion

It has been well characterized that many genes such as kinases, disease resistance (R) proteins, pathogenesis-related (PR) proteins, and hormone-related genes play key roles in response to pathogen attack through activating plant defense systems [46]. Functions of PR genes in response to biotic or abiotic stresses, especially in responding to various types of pathogens, have already been well characterized in plants [47]. In our study, we found that 15 PR genes were differentially expressed, and most of them were up-regulated at 24 hpi in VI037621 post *Xcv* infection (Appendix A and Appendix A). For instance, PR gene Capana03g004441 and Capana01g000279 were up-regulated at 24 hpi, while Capana09g002059 was down-regulated at 24 hpi. As the receptors of pathogen signals, most of the 57 pattern recognition receptors (PRRs) were up-regulated post *Xcv* infection (Appendix A and Appendix A). Mitogen-activated protein kinases (MAPKs), calcium signaling genes can function in the signal transduction processes (Appendix A, and Appendix A). According to functional annotations of these DEGs homologs in *Arabidopsis thaliana* (Appendix A), we found that the majority of them were associated with plant defenses or stress responses. For instance, *OSM34* in *Arabidopsis*, corresponding to the homologous gene Capana01g000279 in pepper, has been found to induce a defense response to the ochratoxin A-producing strain [48], while *AtPRB1*, corresponding to the homologous gene Capana09g002059 in pepper, encodes a basic PR1-like protein and responds to ethylene and methyl jasmonate, and its expression can be suppressed by salicylic acid (SA) [49]. In addition, many genes encoding disease R proteins were highly expressed in VI037601 at three time points post *Xcv* infection, especially at 6 hpi (Appendix A, Appendix A). Thus, all of these genes are essential for enhancing the host against biotic infection.

Plant hormones, such as auxins (IAA), jasmonic acid (JA), SA, and ethylene, have been reported to act as key regulators during the entire course of the plant development and response to internal and external environmental cues [50]. For instance, JA has been found to involve in defense reactions to necrotrophs and induced systemic resistance with ethylene [51]. In our study, all differentially expressed JA-related genes were down-regulated at 24 hpi in VI037601 (Appendix A and Appendix A). SA typically mediates basal defenses to biotrophic pathogens and is essential for the rapid activation of local and systemic resistance [52]. Several genes encoding salicylic acid-binding protein like Capana00g003613 were also up-regulated in VI037601 (Appendix A and Appendix A). These results indicate that the hormone signaling pathways such as JA and SA are also involved in the *Xcv* defense of pepper.

RNA-seq data have been widely applied to identify DEGs related to tissue-, developmental stage-, or stress response-specific expression in plant species [53]. It is a widely accepted concept that the aforementioned DEGs are too numerous to find key genes involved in the specific traits we examined. Therefore, it is necessary to develop and utilize other approaches helping to narrow down the DEGs to several candidates for validation. WGCNA is an effective method that has been widely applied to mine key Hub genes involved in some specific biological processes in plants [21,54,55,56]. In this study, through WGCNA, we identified 15 Hub TFs from three modules related to *Xcv* resistance response in pepper (Figure 3B–D), thereby significantly increasing efficiency to identify potential Hub genes responsible for *Xcv* resistance response in pepper. 

However, WGCNA usually cannot provide enough evidence to predict the regulatory relationships between co-expressed genes. Fortunately, software such as ARACNE [25] and GENIE3 [26] could predict regulatory networks based on co-expression networks [24]. For example, through combining WGCNA and GENIE3, some key TFs in response to the low nitrogen in rice were successfully predicted and experimentally validated [31], confirming the accuracy of these methods in mining key genes related to some agronomic traits of interest. To examine the regulatory relationships between TFs or between TFs and their target genes, we further employed a machine learning approach to infer regulatory causality for co-expressed genes, and 20 TFs were selected as candidates. In agreement, we predicted 4 TFs as key regulators involved in BS disease resistance in pepper (Figure 5A), further indicating the power of WGCNA in combination with GENIE3 in the identification of key genes responsible for agronomic traits.

It has been reported that *AtHSFA6B*, corresponding to the homologous gene Capana03g003818 we predicted in pepper, is a positive regulator involved in abscisic acid-related stresses, such as salt, drought, and heat tolerance [57], suggesting that Capana03g003818 may function in *Xcv* resistance in pepper. Moreover, *AtCAR1* in *Arabidopsis*, corresponding to the homologous gene Capana09g000015 in pepper, which connected module pink and black, is a disease-resistant protein containing the NB-ARC domain that recognizes conserved effectors *AvrE* and *HopAA1* [58], therefore, suggesting potential roles of Capana09g000015 in *Xcv* resistance in pepper. It has been found that overexpression of *DREB2C* in *Arabidopsis*, corresponding to the homologous gene Capana00g004634 in pepper, can enhance heat tolerance [59]. The *Arabidopsis BLH1*, corresponding to the homologous gene Capana11g000364 in pepper, has been reported to regulate seed germination and seedling development [60], suggesting possible roles of Capana11g000364 in some fundamental biological processes. 

In addition to the direct involvement of these 4 TFs in *Xcv* resistance and development in pepper, our study further provided evidence to show that these TFs can be indirectly involved in *Xcv* resistance by regulating the expression of some key genes (Appendix A). For example, some of these common target genes of 3 TFs in the black module were related to chromatin and its modifications, such as Capana00g000428 and Capana05g002474 for encoding Histone H2A and H2B, respectively, Capana12g000665 for encoding Histone-lysine n-methyltransferase 2d-like, and Capana05g002192 and Capana04g000356 for encoding putative methyltransferase pmt11 and pmt28, respectively, further implying that involvement of these TFs in response to *Xcv* infection in pepper may be mediated by chromatin dynamics. It is noted that the roles of these 4 TFs in *Xcv* resistance need to be further experimentally validated in the future. 

It has been predicted that 21 TFs such as MYB, ERF, and WRKY were specific differentially expressed in VI037601 post *Xcv* infection; among 63 TFs differentially expressed in ECW and VI037601 post *Xcv* infection, up-regulation of MYB, WRKY, ERF, HSF, and bHLH TFs was detected in both cultivars post *Xcv* infection; those TFs are potentially candidates responsible for resistance to *Xcv* infection [32]. In contrast, it turns out that the 4 TF candidates, Capana00g004634 (ERF family), Capana09g000015 (C3H family), Capana11g000364 (TALE family), and Capana03g003818 (HSF family) in our study are different from the aforementioned differentially expressed TFs in the original paper [32] even though they were all considered as candidate genes related to *Xcv* resistance in pepper in the two different studies. In our opinion, different conclusions regarding candidates of resistance genes between our study and the original paper [32] could reflect variations in outcomes caused by the application of different prediction methodologies. In the original article [32], the candidates of resistance genes were predicted based on DEGs that occurred in ECW and VI037601 at the time points (6hpi and 24 hpi) post *Xcv* infection, whereas candidates of resistance genes were predicted in our study through WGCNA in combination with GENIE3 network constructed using almost all genes instead of DEGs. Both results could be complementary with each other, thereby providing more candidates for pepper breeding even though the prediction accuracy still awaits future validation.

## 5. Conclusions

Through the association analysis of WGCNA and GRN, we predicted 4 TF candidates involved in the processes of *Xcv* resistance response in pepper, especially Capana09g000015. We further showed that these TFs coordinated to function in *Xcv* resistance response through regulating their key target genes. Thus, our study provides valuable references for identifying key candidates for disease resistance in horticulture crops, thereby helping to elucidate the underlying molecular mechanism responsible for *Xcv* resistance in pepper. 

## Figures and Tables

**Figure 1 genes-12-01351-f001:**
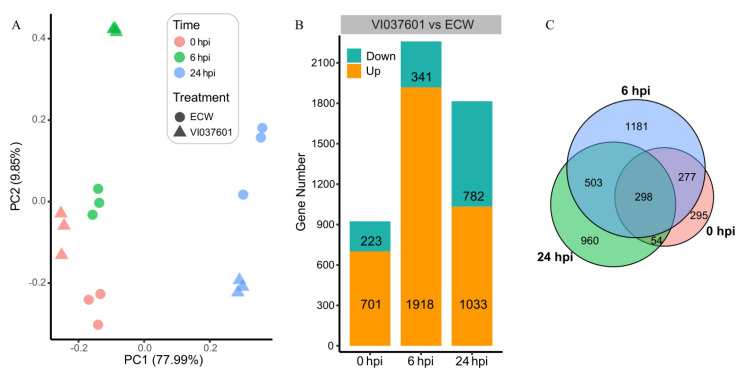
Transcriptional changes occurred between VI037601 and ECW post *Xcv* infection. (**A**) Principal component analysis (PCA) of all 18 RNA-seq samples (three biological replicates per time point) generated from VI037601 and ECW at different time points (0, 6, and 24 hpi) post *Xcv* infection. Distance between/among samples for PC1 or PC2 directly corresponds to the correlation between/among samples examined. (**B**) Statistics of differentially expressed genes (DEGs) numbers in VI037601 relative to ECW at three time points (0, 6, and 24 hpi) post *Xcv* infection. The green color represents down-regulated genes, and the yellow color represents up-regulated genes in VI037601 relative to ECW occurred at each time point post *Xcv* infection. (**C**) The Venn plot showing pairwise comparisons of total DEGs that occurred at each time point in VI037601 relative to ECW post *Xcv* infection. For instance, 298 DEGs were shared at three time points post *Xcv* infection; 295, 1181, and 960 DEGs specifically occurred at 0, 6, and 24 hpi, respectively, post *Xcv* infection.

**Figure 2 genes-12-01351-f002:**
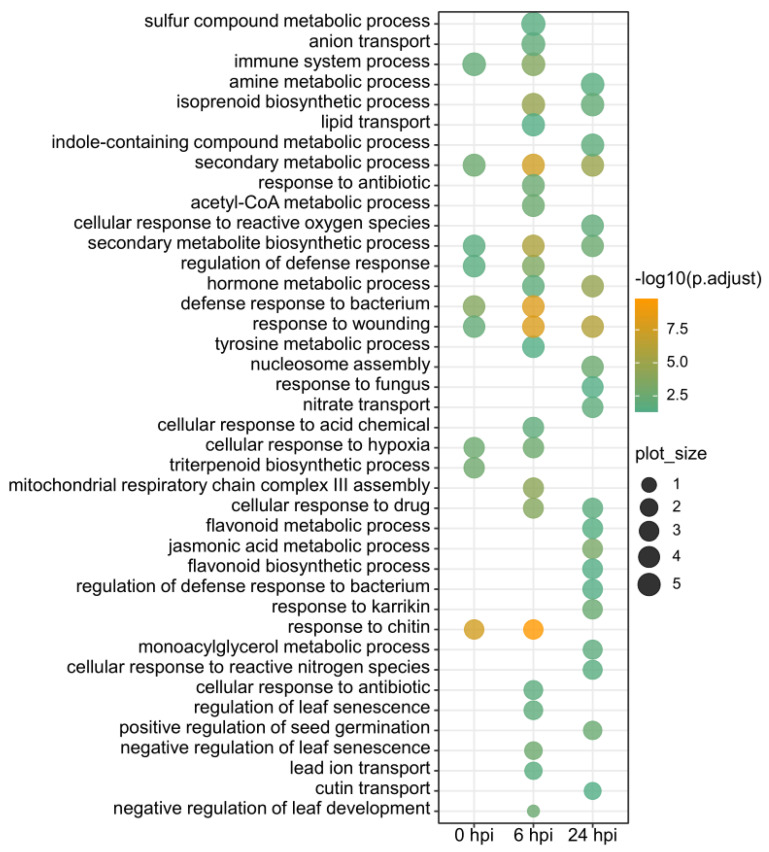
GO functional enrichment analysis of DEGs between VI037601 and ECW at three time points (0, 6, and 24 hpi) post *Xcv* infection. Significant common or distinct GO terms were obtained for DGEs across three time points or at each time point. For instance, secondary metabolic processes and response to wounding GO terms were common for DEGs at three time points, while response to chitin or hypoxia GO terms were detected in DEGs occurring at 0 and 6 hpi.

**Figure 3 genes-12-01351-f003:**
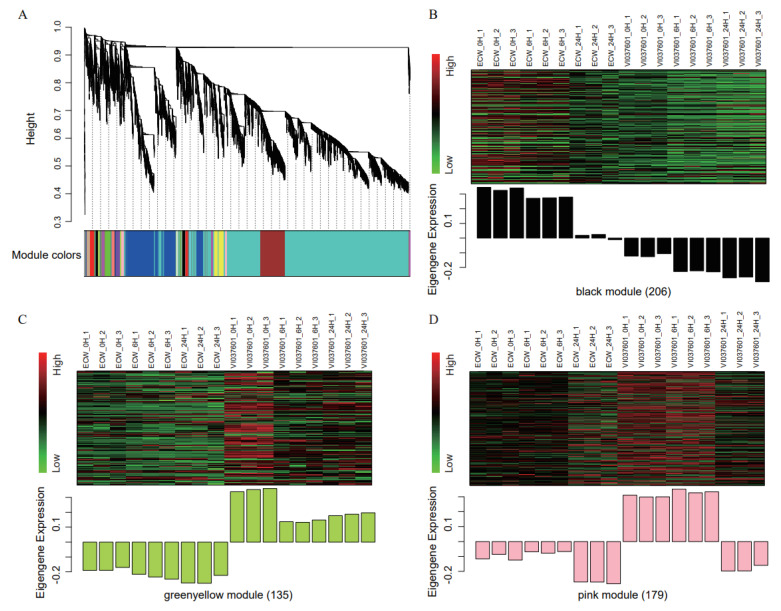
Construction and identification of co-expression modules. (**A**) Detection of gene cluster dendrograms and modules. Hierarchical clustering tree (dendrogram) showing coexpression modules identified via the Dynamic Tree Cut method. The major tree branches constitute 18 modules which are labeled with different colors. The genes were clustered on the basis of dissimilarity measure (1-TOM). The branches correspond to modules of highly interconnected genes, each ‘leaf’ (short vertical line) in the tree represents one individual gene. (**B**–**D**) Heatmaps showing the expression profiles of co-expressed genes from module black (**B**), greenyellow (**C**), and pink (**D**), bar graphs (below the heatmaps) showing the consensus expression pattern of the coexpressed genes in each module. Red, black, and green represent high, medium, and low expressions, respectively. The number in parentheses (labeled on bottom) represents the number of genes in the module.

**Figure 4 genes-12-01351-f004:**
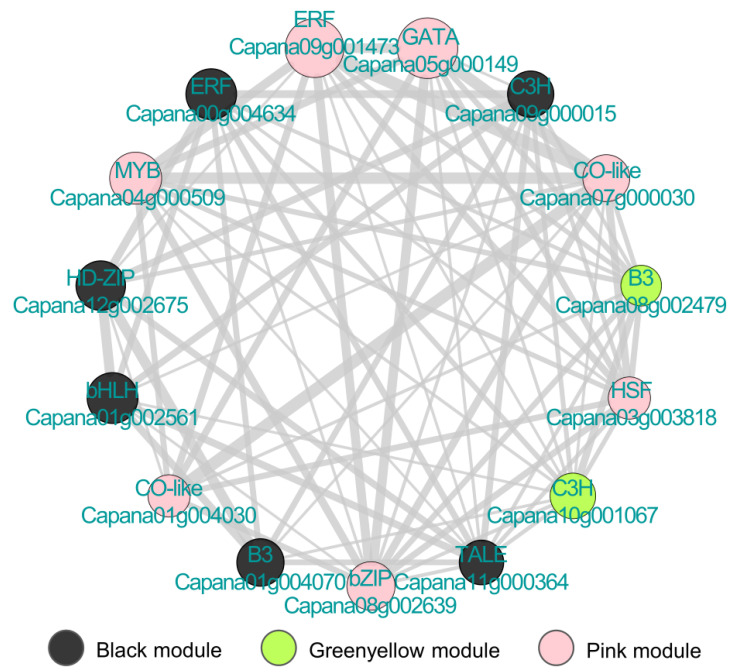
Identification of Hub TFs in three modules related to *Xcv* resistance response. Hub TF network was derived from three key co-expression modules. Pearson’s correlation coefficient between all genes vs. module eigengene expression was calculated, which was defined as ‘kME’. Genes with |kME| > 0.85 were defined as Hub genes. Hub TF genes of module black, pink, and greenyellow are indicated using black, pink, and greenyellow color points, respectively. Correlations between the expression of Hub TF genes are shown with grey lines. The thickness of the line indicates the strength of the co-expression intensity, and the size of the point indicates the level of |kME|.

**Figure 5 genes-12-01351-f005:**
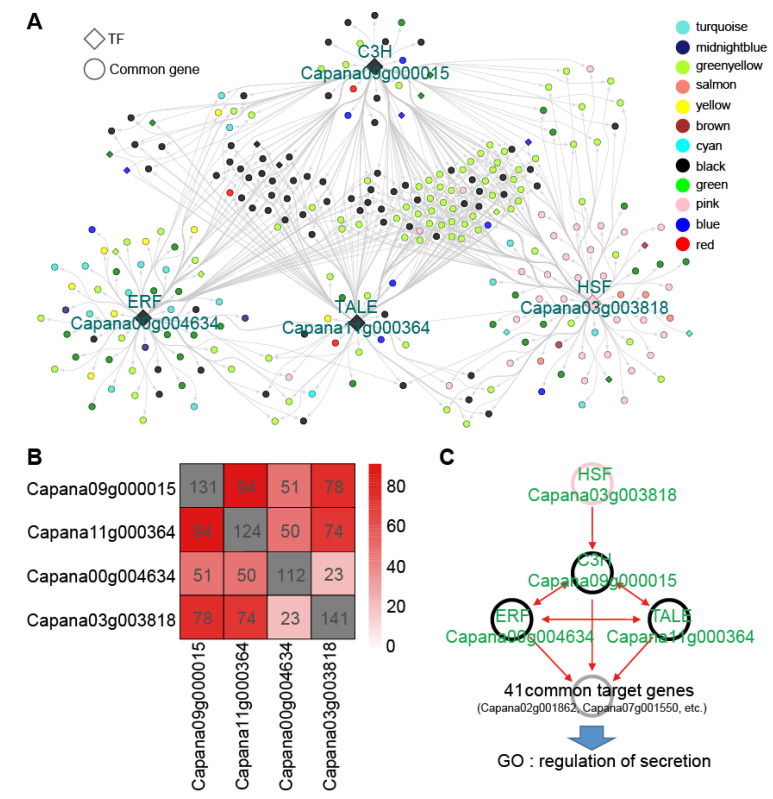
Identification of key TFs and their target genes in GRN. (**A**) The transcriptional regulatory sub-network of 4 TF candidates. Overview of the high-confidence gene regulatory sub-network. Each node represents a gene, and genes exhibiting regulatory relationships are connected with edges. The gray arrow indicates the direction of gene regulation. The diamond frames represent TFs, while the circles represent other genes. Genes belonging to different co-expression modules (as determined by WGCNA) are indicated in different colors that correspond to those of co-expression modules. The same colors have been used in Figure 3 and Figure 5, and Appendix A to indicate genes belonging to the same co-expression modules. (**B**) Overlaps among the predicted target genes of 4 TF candidates in GRN. Genes connected with a transcription factor among the top 10,000 edges were defined as putative targets. The numbers in the gray squares represent predicted TF target genes. (**C**) Schematic representation of mutual regulation relationships of TF candidates and the putative common target genes of 3 TFs in the black module. The red arrow indicates the direction of gene regulation. TFs belonging to module pink and black are marked using pink and black circles, respectively. The grey circle indicates 41 common target genes of 3 TFs in the black module. Genes with the GO term ‘regulation of secretion’ were enriched among the common target genes of 3 TFs in the black module.

**Figure 6 genes-12-01351-f006:**
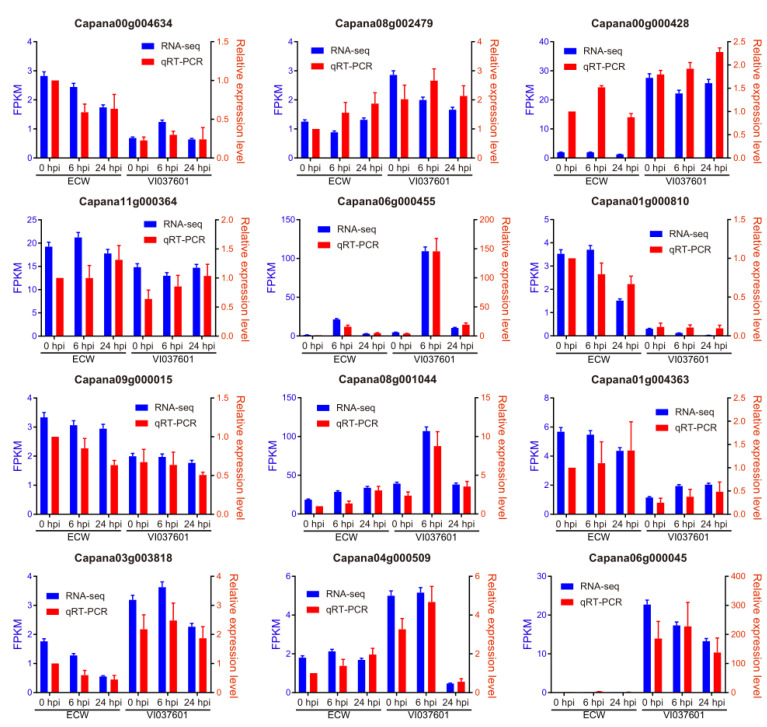
qRT-PCR validation of some key genes identified in the network. A dual y-axis plot illustrating an in-parallel comparison between RNA-seq (FPKM, blue) and RT-qPCR (relative expression levels, red) of each individual gene examined. The left coordinate (y-axis) with blue represents the FPKM value of RNA-seq, the right coordinate (y-axis) with red represents the relative expression levels of qRT-PCR. The transcript levels of each gene at each time point were normalized relative to the internal control of gene *CaUbi3*. Relative expression levels of genes examined were calculated and expressed as 2^−^^ΔΔ^^CT^ relative to the expression levels of the corresponding genes in ECW at 0 hpi, which were set as 1.0. The mean expression levels were calculated from three biological replicates. Error bars are the standard deviations of three biological replicates.

**Table 1 genes-12-01351-t001:** Motif prediction of the target genes of 4 TFs.

TF Candidates	Motifs	*p*-Value	E-Value	Known TF ID
TALE Capana11g000364	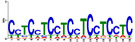	1.20 × 10^−20^	1.04 × 10^−17^	TF3A
	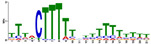	4.79× 10^−15^	4.18 × 10^−12^	OBP3
	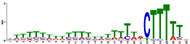	4.97 × 10^−14^	4.33 × 10^−11^	At4g38000
HSF Capana03g003818	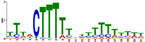	3.42 × 10^−17^	2.98 × 10^−14^	OBP3
	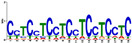	1.69 × 10^−14^	1.47 × 10^−11^	TF3A
	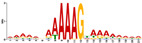	4.32 × 10^−13^	3.76 × 10^−10^	Adof1
ERF Capana00g004634	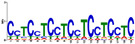	7.03 × 10^−18^	6.13 × 10^−15^	TF3A
	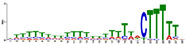	1.88 × 10^−13^	1.64 × 10^−10^	At4g38000
	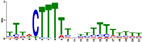	5.66 × 10^−13^	4.93 × 10^−10^	OBP3
C3H Capana09g000015	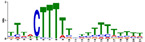	3.18 × 10^−15^	2.78 × 10^−12^	OBP3
	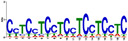	1.50 × 10^−13^	1.31 × 10^−10^	TF3A
	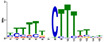	1.89 × 10^−12^	1.65 × 10^−9^	AT1G47655

## Data Availability

The data presented in this study are available in the article and Appendix A.

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
