# Peer review of "Identification of Key Transcription Factors Related to Bacterial Spot Resistance in Pepper through Regulatory Network Analyses"

_genes, 2021, doi:10.3390/genes12091351_

Round 1

Reviewer 1 Report

Zhu and Zhang have compiled this study and have tried to utilize the public available RNA seq data sets upon Xanthomonas campestris pv. Vesicatoria infection and analyzed the TFs and their GRNs.

Though this study has provided some new information about the transcription factor and its target genes co expression, the overall novelty and presentation could be greatly improved, below my comments for the updated version.

  1. I cannot understand the motivation from the introduction for using VI037601 and ECW cultivars differently in analyzing the datasets. I would rather add an extra analysis in looking how the behavior of the genes differ with in the cultivars, infact  this can provide more information to the readers, in addition to the presented ones.
  2. In the figure 2 authors have used the GO term enrichment, what is puzzling to me is 24h time point, expected changes such as response to bacterium is also very modest, how authors can explain this?, is this an artefact of using large data sets that are not quite well pre processed?
  3. I will find or highly recommend to simplify the fig 3A, what is the module colours vs genes correlation is not clear from the text, Scale missing for the #3B-D
  4. Can authors find the target genes promoter regions and looks for the cis element which can be good evidence for the direct binding, although authors have provided the target gene expression, still it can be indirect and has very little to do with the direct binding.
  5. Can authors employ ChIP SEQ datasets if available for the pepper and see how is the correlation behaving for the TF genes of interest in f5
  6. Please write the figure legends more descriptively, the important information should be given clearly for the readers to repeat the work, because the only new information for these type of works is the way the data has been analyzed, if this is not clear to the reader’s manuscript makes little sense.
  7. What are the up and down regulated genes homologs in Arabidopsis, a table will be helpful and this has to be well discussed in discussion, current discussion is weak in giving new information

minor

Abbreviate, TALE and other TF families

Use size based/proportional venn diagram

Please explain in the results what the cytoscape figure depicts, its not clear

Are authors depositing the R codes to Git hub as standard practice, if yes can they provide the access to the update version of the manuscript    

Author Response

Response to reviewer 1 comments for genes-1333683

Comments and Suggestions for Authors

Zhu and Zhang have compiled this study and have tried to utilize the public available RNA seq data sets upon Xanthomonas campestris pv. Vesicatoria infection and analyzed the TFs and their GRNs. Though this study has provided some new information about the transcription factor and its target genes co expression, the overall novelty and presentation could be greatly improved, below my comments for the updated version.

Response: We appreciate the reviewer’s affirmation on our work. Especially, we appreciate the time the reviewer spent making these constructive comments on the manuscript. These comments are all valuable and helpful for revising and improving our manuscript.

  1. I cannot understand the motivation from the introduction for using VI037601 and ECW cultivars differently in analyzing the datasets. I would rather add an extra analysis in looking how the behavior of the genes differ with in the cultivars, in fact this can provide more information to the readers, in addition to the presented ones.

Response: We gratefully appreciate your valuable comment. We apologize for unclear descriptions of our motivation for conducting related analyses in the introduction. The differentially expressed genes (DEGs) in VI037601 or early Calwonder (ECW), representing resistance and susceptible to Xcv infection, respectively, at 0h, 6h and 24h post Xcv infection, have already been reported by Gao et al. (PLOS One (2021) 6, e0240279), whereas it is unclear how genes are differentially expressed between VI037601 and ECW at 0h, 6h and 24h post Xcv infection. Thus, we mainly focused on identification of DEGs occurred between VI037601 and ECW at 0h, 6h and 24h post Xcv infection and conducted regulatory network analyses for predicting key TFs involved in response to Xcv infection in pepper. We added some descriptions to introduce our motivation for initiating some related analyses in this study.

Following the suggestion, we reanalyzed the published time-point related RNA-seq datasets for identifying DEGs occurred in VI037601 or ECW at each time point post Xcv infection (see Figure 1 and 2 below). We found that they are overall similar to those in the original paper (PLOS One (2021) 6, e0240279). To avoid repeating the results already published, we did not add those results in the text.

Figure 1. Identification of differentially expressed genes (DEGs) in ECW and VI037601 leaves at 6 hours and 24 hours post Xcv infection. (A) Numbers of up- and down-regulated DEGs at 6 hpi and 24 hpi in ECW and VI037601, respectively. (B) Venn diagram showing common and unique DEGs in ECW_6H-vs-0H, ECW_24H-vs-0H, VI037601_6H-vs-0H and VI037601_24H-vs-0H.

Figure 2. GO enrichment analyses of DEGs in ECW and VI037601 at 6 hours and 24 hours post Xcv infection. The color scale represents significance (corrected P-value). Each point represents the number of genes enriched in GO terms.

  1. In the figure 2 authors have used the GO term enrichment, what is puzzling to me is 24h time point, expected changes such as response to bacterium is also very modest, how authors can explain this? is this an artefact of using large data sets that are not quite well pre processed?

Response: We thank the reviewer very much for pointing out this problem. In our opinion, differential enrichment of GO terms associated with response to bacterium between 6 hpi and 24 hpi is less likely caused by data pre-processing. The plausible explanation: 6 hpi is the key time point for VI037601 and ECW responding to Xcv infection through up- or down-regulating some of genes functioning in defense response and in response to antibiotic, bacterium, chitin, hypoxia, wounding, acid chemical and drug, which are essential steps toward enhancing resistance to Xcv infection in VI037601, thereby reducing disease-related response and promoting recovery of plants for normal growth for some extent at 24 hpi. This process may cause differential enrichment of GO terms associated with stress or bacterium responses at 24 hpi relative to 6 hpi. In addition, the GO term related to 'regulation of defense response to bacterium' was relatively weak for DEGs at 24 hpi, but it was still statistically significant, and other GO terms primarily involved in defence or stress such as response to karrikin, jasmoni acid metabolic process and hormone metabolic process were also enriched for DEGs at 24 hpi. Moreover, some GO terms associated with metabolic and synthesis-related were specifically enriched for DEGs at 24 hpi. Those DEGs with significantly enriched GO terms mentioned above may function in stress response and some fundamental developmental processes, thereby resulting in infection-related phenotypic changes between two cultivars at 24 hpi (see Figure 1 in the published paper below).

We accordingly modified descriptions in the text.

  1. I will find or highly recommend to simplify the fig 3A, what is the module colours vs genes correlation is not clear from the text, Scale missing for the #3B-D

Response: We thank the reviewer very much for pointing this out. We are sorry for no description about the module colors vs gene correlation in the text. Figure 3A is the gene cluster dendrograms, which is the original figure of the clustering results generated by the software WGCNA when building the co-expression modules. Gene clustering is performed according to the correlation between genes. To show correlation between the module colors and genes, we added a supplemental Table S3. In addition, we added the legend bar for the figure 3B-D as you suggested.

  1. Can authors find the target genes promoter regions and looks for the cis element which can be good evidence for the direct binding, although authors have provided the target gene expression, still it can be indirect and has very little to do with the direct binding.

Response: We appreciate the reviewer’s great comment. Following the suggestion, we extracted DNA sequences located at 1 kb upstream of the TSS of 4TF targeting genes identified through the network for motif identification using AME. We found that CCTCCT-repeat and CTTTTT motifs were enriched in the promoters of 4 TF targeting genes. We listed the results in Table 1 and added the description as “3.5. Validation of TF candidates” in the text (line 305 to 315). 

  1. Can authors employ ChIP SEQ datasets if available for the pepper and see how is the correlation behaving for the TF genes of interest in f5.

Response: We appreciate the reviewer’s great comment, which is helpful for advancing our understanding of how TF genes of interest function in disease resistance in pepper. We completely agree with you. However, unfortunately we did not find any TF-related ChIP-seq datasets available for further analyses.

  1. Please write the figure legends more descriptively, the important information should be given clearly for the readers to repeat the work, because the only new information for these type of works is the way the data has been analyzed, if this is not clear to the reader’s manuscript makes little sense.

Response: We apologize for unclear description of figure legends. Following suggestions, we added more details in each figure legend.

  1. What are the up and down regulated genes homologs in Arabidopsis, a table will be helpful and this has to be well discussed in discussion, current discussion is weak in giving new information

Response: Thank you for your keen insight and high-level comment, which help us to ameliorate the manuscript again. Following the suggestion, we have listed all up and down regulated genes homologs in Arabidopsis in Supplementary Table S1. Accordingly, we added more discussion regarding functions of some homologs well characterized in Arabidopsis in the discussion section.

minor

Abbreviate, TALE and other TF families

Response: Following the suggestion, we added the full name for TALE and other TF family in the text. For example, TALE represents the three-amino acid loop extension; HSF represents heat shock transcription factor. Thank you!  

Use size based/proportional venn diagram

Response: Following the suggestion, Figure 1C has been replaced by the proportional venn diagram. Thank you!

Please explain in the results what the cytoscape figure depicts, its not clear

Response: We apologized for the inconvenience ensued. We added more descriptions for the legend of cytoscape figures.

Are authors depositing the R codes to Git hub as standard practice, if yes can they provide the access to the update version of the manuscript    

Response: We exactly followed the official tutorials for WGCNA and GENIE3. We added the codes we used as a supplementary file Methods S1 as suggested. Thank you!

Reviewer 2 Report

The present study would like to reanalyze the public multiple time-point related RNA- seq data sets generated from two pepper cultivars, the Xcv-susceptible cultivar ECW and the Xcv-resistant cultivar VI037601, post Xcv infection from published article (PLOS One (2021) 6, e0240279). They try to identify the TFs and their regulatory target genes responsible for Xcv resistance through co-expression and regulatory networks in pepper. They used WGCNA combined GENIE3 analysis to cluster possible key TFs hub and co-expressed TFs with respect to Xcv resistance. This ms didn’t compared their result with original paper published. They should explain why most of TFs hubs selected in this study are different than original paper. Our other concerns as following:

  1. 278-280, The authors announced that 4 TFs as key regulators involved in BS disease resistance in pepper (Figure 5A), further indicating the power of WGCNA in combination with GENIE3 in identification of key genes responsible for agronomic traits. However, no enough evidence to support their inference. All their inference didn’t provide any real-time PCR or RT-PCR evidence.
  2. .80-92, the authors almost copy whole paragraph from PLOS One (2021) 6, e0240279. Is it ok?
  3. Although the authors acknowledged Drs. Shenghua Gao, Chunhai Jiao and Minghua Yao (first author and corresponding authors of PLOS One, 2021) for providing the annotation information of Zunla-1 pepper genome. We still concerned is it suitable to re-analyze a single published data set and may provide different conclusion compared to original paper.
  4. The references format should be unified. Please check all the references.

Round 2

Reviewer 1 Report

Authors have significantly answered by queries, 

I have a small suggestion for figure 6, I would recommend using the standard bar graphs for the qRT-PCR with the data points impeeded on it, and compare it with the RNA. I am puzzled how do they managed to normalize the scale between these two, I would also appreciate it if the authors describe this in the legend or in the text, 

as an additional note, can authors have detailed legends in the final version, 

thank you, the manuscript overall is fit for publication. 

Reviewer 2 Report

The revised ms have included Dr. Gao as the co-authors and provided more detail description and comparison with earlier reference. They have suitable responses for all reviewed concerns.